# Visible Light Promotes Cascade Trifluoromethylation/Cyclization, Leading to Trifluoromethylated Polycyclic Quinazolinones, Benzimidazoles and Indoles

**DOI:** 10.3390/molecules27238389

**Published:** 2022-12-01

**Authors:** Ransong Ma, Yuanyuan Ren, Zhoubin Deng, Ke-Hu Wang, Junjiao Wang, Danfeng Huang, Yulai Hu, Xiaobo Lv

**Affiliations:** 1College of Chemistry and Chemical Engineering, Northwest Normal University, Lanzhou 730070, China; 2State Key Laboratory of Applied Organic Chemistry, Lanzhou University, Lanzhou 730000, China; 3Shanghai Sinofluoro Chemicals Co., Ltd., Shanghai 201321, China

**Keywords:** bromotrifluoromethane, visible-light induction, trifluoromethylation, cyclization, polycyclic quinazolinone, polycyclic benzimidazole, polycyclic indole

## Abstract

Efficient visible-light-induced radical cascade trifluoromethylation/cyclization of inactivated alkenes with CF_3_Br, which is a nonhygroscopic, noncorrosive, cheap and industrially abundant chemical, was developed in this work, producing trifluoromethyl polycyclic quinazolinones, benzimidazoles and indoles under mild reaction conditions. The method features wide functional group compatibility and a broad substrate scope, offering a facile strategy to pharmaceutically produce valuable CF_3_-containing polycyclic *aza*-heterocycles.

## 1. Introduction

Organofluorides are ubiquitous in pharmaceuticals, agrochemicals and functional materials owing to their high lipophilicity, improved metabolic stability and enhanced bioactivity compared with their parent molecules [1,2,3,4,5,6]. Until now, more than 340 fluorinated pharmaceuticals and 420 fluorinated agrochemicals have been registered and used commercially, which account for 20% of clinical drugs and 35% of commercial agrochemicals, respectively [7,8]. Thus, organofluorides have become increasingly important in developing new pharmaceuticals and agrochemicals. Among the various organofluorides, trifluoromethyl compounds are of great importance due to their frequent appearance in drug and pesticide molecules. This fact has stimulated intensive investigations of trifluoromethylation in organic compounds, and various trifluoromethylation methodologies have been developed. In particular, the appearance of versatile trifluoromethylation reagents such as the Togni [9,10,11], Umemoto [12], Ruppert–Prakash [13,14] and Langlois [15] reagents [16,17] have greatly pushed forward the development of trifluoromethylation methodologies through electrophilic, nucleophilic and radical processes. However, all of these reagents are expensive, and none of them are commercially available in bulk quantities for the time-being. Hence, the exploration of new cost-effective and atom-economic trifluoromethylation methods using easily available trifluoromethylation reagents in bulk quantities in industry is of great value. To address this question, cost-effective and readily available trifluoroacetic acid (TFA), trifluoroacetic anhydride (TFAA) and triflic anhydride (Tf_2_O) have recently been investigated as trifluoromethylation reagents under diverse catalysis by Zhang, Stephenson, Qing and Ritter, etc. [18,19,20,21,22,23,24,25,26,27,28,29].

As an extinguishant and refrigerant (R-13B1), CF_3_Br is available in large quantities at a low price (21 USD/kg) in industry. Although CF_3_Br has been tested as a trifluoromethylation reagent before, it has not been fully investigated in the trifluoromethylation process compared to other trifluoromethylation reagents. Screening of the literature shows that CF_3_Br can be reduced to a trifluoromethyl anion via electrochemical reduction [30,31], active metal reduction [32,33], or by P(NEt_2_)_3_ [34] to react with aldehydes or ketones to give alcohols, or to react with TMSCl to give TMSCF_3_ (Figure 1a). Secondly, CF_3_Br can be reduced to a trifluoromethyl radical by Na_2_S_2_O_4_ [35] or the transition metal complex [36,37,38,39] to couple with aromatic rings or alkenes to give trifluoromethyl compounds (Figure 1b). In 2018, Zhang′s group reported that CF_3_Br could be activated by visible-light catalysis to produce a trifluoromethyl radical and induce a radical addition reaction with alkenes and alkynes (Figure 1c) [40]. This process features an environmentally friendly and sustainable strategy to activate CF_3_Br. In our previous work, the visible-light-induced trifluoromethylation of *O*-silyl enol ether was initially tested using CF_3_Br as trifluoromethyl reagent under visible-light catalysis (Figure 1d) [41]. As our research on CF_3_Br is on-going, we herein report a visible-light-induced radical cascade trifluoromethylation/cyclization reaction, which provides a series of trifluoromethylated polycyclic quinazolinones, benzimidazoles and indoles, using CF_3_Br as a trifluoromethyl source (Figure 1e). This transformation provides a facile way to construct polycyclic *aza*-heterocycles, which are usually found in various pharmaceutical compounds (Figure 2). Moreover, this approach features a low-cost trifluoromethyl source and wide substrate tolerance compared with other methods [42,43,44,45,46,47,48].

## 2. Results and Discussion

The initial investigation commenced with the reaction of 3-(pent-4-en-1-yl)quinazoline-4(3H)-one (**1a**) and CF_3_Br (**2**) in the presence of tris(2-phenylpyridine)iridium (*fac*-Ir^III^(ppy)_3_) in acetonitrile (CH_3_CN) under the irradiation of a 5 W blue LED (460–465 nm). The anticipated product **3a** was obtained with only a 5% yield (Table 1, entry 1). In view of the great influence of solvent effects on the reaction, various commonly used solvents including dichloromethane (DCM), toluene (Tol), tetrahydrofuran (THF), dimethyl sulfoxide (DMSO), *N, N*-dimethylformamide (DMF) and *N*-methyl pyrrolidone (NMP) were tested; NMP proved to be the most suitable solvent, in which the product **3a** was obtained in a 30% yield (Table 1, entries 2–7). Subsequently, the screening of different light sources and their strength showed that the 5 W blue LED was optimal for the reaction (Table 1, entries 8–10). To further improve the product yield, several Lewis acids were examined. Notably, the addition of LiCl (1.0 equiv.) as an additive significantly accelerated the reaction rate, and the desired product **3a** was obtained in an 83% yield (Table 1, entries 11–15). Afterwards, the screening of the loading of the photocatalyst (PC) indicated that 1 mol % of *fac*-Ir^III^(ppy)_3_ was still the most suitable amount (Table 1, entries 14, 16 and 17). Finally, control experiments showed that a photocatalyst or light source is necessary for this transformation because there was no product formed in the absence of the photocatalyst or blue LED light (Table 1, entries 18 and 19). Thus, the optimal reaction conditions were to perform the reaction with 1 equiv. of 3-(pent-4-en-1-yl)quinazoline-4(3H)-one, under 1 atm CF_3_Br, in the presence of *fac*-Ir^III^(ppy)_3_ in NMP, under the irradiation of a 5 W blue LED, with 1 equiv. of LiCl as the additive.

With the optimal reaction conditions in hand, the substrate scopes were investigated. First, various *N*-alkenyl quinazolinones **1** were examined (Figure 3). The results indicated that substrates containing both electron-donating (methyl and methoxy) and electron-withdrawing groups (fluoro-, chloro- and trifluoromethyl) at the 5-, 6-, 7- or 8-position of the quinazolinone ring were well tolerated and provided the corresponding ring-fused quinazolinones in 30% to 73% yields (Figure 3, **3b**–**3o**). *N*-alkenyl quinazolinones possessing disubstituted benzene rings reacted well to generate the desired products **3p**–**3r** in 32−81% yields (Figure 3, **3p**–**3r**). In addition, five- and seven-membered cyclized products **3s**–**3v** were acquired in 30−76% yields (Figure 3, **3s**–**3v**). Finally, when the benzene ring of the quinazolinone was replaced by a pyridine or thiophene moiety, the corresponding products **3w** and **3x** were isolated in 41% and 69% yields (Figure 3, **3w** and **3x**).

Next, various *N*-alkenyl pyrroles and *N*-alkenyl indoles **4** were applied in the standard conditions. As illustrated in Figure 4, *N*-alkenyl pyrroles with acetyl groups at the 2-position gave five- and six-membered cyclization products (Figure 4, **5a** and **5b**). However, pyrroles with methyl at the 2-position and without substituents failed to produce the desired product (Figure 4, **5c** and **5d**). These results indicated that electron-withdrawing substitution on the pyrrole was favorable to the reaction. Afterwards, *N*-alkenyl indoles containing different substituents were also examined in this transformation. Similarly, indoles with electron-withdrawing substituents (e.g., Ac, CF_3_CO and CO_2_Me) at the 3-position produced the desired products (Figure 4, **5e**–**5h**), while no desired product was detected when the R group at the 3-position was H or methyl (Figure 4, **5i** and **5j**).

In order to further investigate the substrate generalities, various *N*-alkenyl benzimidazoles **6** were applied to the reaction (Figure 5). When 1-(pent-4-en-1-yl)-1H-benzimidazole reacted with CF_3_Br under the standard reaction conditions, the six-membered cyclic product **7a** was produced in a 70% yield. The other derivatives of 1-(pent-4-en-1-yl)-1H-benzimidazole with both electron-donating and electron-withdrawing groups on their phenyl rings also produced the corresponding products in good yields (Figure 5, **7b**–**7f**). Subsequently, the reaction was extended to linear and branched *N*-butenyl benzimidazole, and the corresponding five-membered cyclization products **7g** and **7h** were also generated smoothly, albeit with decreased yields (Figure 5, **7g** and **7h**). Moreover, 7-azobenzimidazole and theophylline were successfully used in this reaction to give the products **7i** and **7j** with good yields of 62% and 70% (Figure 5, **7i** and **7j**).

## 3. Gram-Scale Synthesis

To probe the practical utility of this trifluoromethylation process, a gram-scale reaction of 3-(pent-4-en-1-yl) quinazolin-4(3H)-one was performed with 1 mol % catalyst loading, which proceeded smoothly to produce the desired product **3a** in a 60% yield (Figure 6). Experimental details and characterization data for products are in Appendix A.

## 4. Proposed Mechanism

To probe the reaction mechanism, the radical scavenger 2,2,6,6-tetramethyl-1-piperidinoxyl (TEMPO, 2.0 equiv.) was added in the reaction of **1a** with CF_3_Br under the standard reaction conditions. The formation of the product **3a** was completely inhibited (Figure 7, a). When radical scavenger 1,1-diphenylethylene (2.0 equiv.) was added to the reaction, product **3a** was not formed; only the trifluoromethyl radical trapping compound (3,3,3-trifluoroprop-1-ene-1,1-diyl)dibenzene **8** as produced in a yield of 64% (Figure 7, b). These results imply that the trifluoromethyl radical was involved as the reactive species in the reaction.

In light of our experimental results and the literature descriptions [47,48], a reaction mechanism is tentatively proposed in Figure 8. Under the activation of LiCl, *N*-alkenyl quinazolinones **1a** convert to more electrophilic lithium-activated *N*-alkenyl quinazolinones **A**. Meanwhile, the visible light induced the transformation of the photocatalyst *fac*-^III^Ir(ppy)_3_ to the excited-state *fac*-^III^Ir(ppy)_3_*, which reduced CF_3_Br to generate a trifluoromethyl radical along with the generation of the *fac*-^IV^Ir(ppy)_3_ complex via a single-electron transfer (SET). Then, the addition of the trifluoromethyl radical onto the C = C bond of lithium-activated *N*-alkenyl quinazolinones **A** gave radical intermediate **B**, which underwent intramolecular radical cyclization to offer the intermediate **C**, followed by a further 1,2-hydrogen shift process to yield the intermediate **D**. The intermediate **D** was then oxidized by *fac*-^IV^Ir(ppy)_3_ to form the cation **E**. Finally, product **3a** was obtained with the loss of a proton.

## 5. Conclusions

In conclusion, we have developed efficient visible-light-induced radical trifluoromethylation/cyclization for the synthesis of potential bioactive trifluoromethylated polycyclic quinazolinones, benzimidazoles and indoles. This system has the advantages of high step-economy and low-cost, which renders this protocol highly attractive for the synthesis of CF_3_-containing polycyclic *aza*-heterocycles.

## Data Availability

Not applicable.

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
