# Peer review of "Visible Light Promotes Cascade Trifluoromethylation/Cyclization, Leading to Trifluoromethylated Polycyclic Quinazolinones, Benzimidazoles and Indoles"

_molecules, 2022, doi:10.3390/molecules27238389_

Round 1

Reviewer 1 Report

The manuscript by Yulai Hu et al. is another extension of the toolbox of trifluoromethylation methods of organic substrates. This visible light induced, iridium photocatalyst assisted, multistep radical reaction of CF3Br with different types of 1,w-azadiene systems provides trifluoroethyl-substituted polycyclic quinazolinones, benzimidazoles and indoles. The study is well executed, the experiments are clearly described and the results are scholarly discussed. A plausible reaction mechanism, oriented on literature precedents, is given. The structure of the products was unambiguously assigned by NMR spectra and HRMS (ESI) and the purity of the products is clearly seen from the copies of 1H and 19F NMR spectra documented in the Supporting Information.

However, before the manuscript is suitable for publication the authors might want to consider a couple of comments.

In the introduction, the authors do not mention published trifluoromethylation reactions using directly fluoroform as reagent. On the other hand, they claim their method as "green" and "atom economic". In my opinion this is an overestimation, having in mind that CF3Br is an ozone depleting compound according to the Montreal protocol and the production of the compound is prohibited in many industrialized countries. Moreover, I am not so sure whether a reaction can be called atom economic when more than 50% of the molecular mass of the reagent is wasted. My suggestion is rephrasing of the corresponding parts over the whole manuscript.

Minor comments

- line 128: H should not be referred to as an electron-donating group, because H is the reference group for the definition of electron-donating or electron-withdrawing properties of substituents. The whole sentence (lines 127-129) is not clear and should be rephrased. I guess that it is meant that no substituent or an electron-donating substituent disfavor the reaction.

line 143: the phrase should read: ... under the standard reaction conditions, the six ...

line 147: the phrase should read: ... the reaction was extended to ...

all Schemes: the arrangement of the reagents and conditions above and below the reaction arrows should be normalized (got out of place)

Scheme 6a: what is the fate of TEMPO in the reaction?

Reviewer 2 Report

Please, see attached pdf-file

Reviewer 3 Report

In this manuscript, the author described a radical cascade avenue to efficiently construct -CF3 contained quinazolinone derivatives. The methodology presents an excellent substrate compatibility and these N- and CF3- containing products are of great interest in medicinal chemistry. The background introduction and presentation are comprehensive, with impressive synthetic efforts, and the mechanism investigation is appropriate to claim the catalytic cycle at this stage. Overall, this is a high-quality manuscript with sufficient scientific significance, this referee therefore support the publication.

Minor suggestions:

1.The background introduction is comprehensive, but I think the specific precedents are not needed to discuss one by one (line 46-60). Select two typical examples to discuss along with other necessary reference numbers should be enough.

2.As per Table 1, the LiCl play a significant role in this transformation compared with other salts. Could the author give some comments for this in the text?
